# Management and Prevention of COVID-19 in Pregnancy and Pandemic Obstetric Care: A Review of Current Practices

**DOI:** 10.3390/healthcare9040467

**Published:** 2021-04-15

**Authors:** Argyro Pountoukidou, Maria Potamiti-Komi, Vrisiis Sarri, Michail Papapanou, Eleni Routsi, Anna Maria Tsiatsiani, Nikolaos Vlahos, Charalampos Siristatidis

**Affiliations:** 1Second Department of Obstetrics and Gynecology, Aretaieion Hospital, Medical School, National and Kapodistrian University of Athens, Vas. Sofias 76, 11528 Athens, Greece; argyropoukd@gmail.com (A.P.); maria.potamiti@gmail.com (M.P.-K.); vrisiida.sarri@hotmail.com (V.S.); mixalhspap13@gmail.com (M.P.); routsie@gmail.com (E.R.); annatsiat@gmail.com (A.M.T.); nikosvlahos@med.uoa.gr (N.V.); 2Assisted Reproduction Unit, Second Department of Obstetrics and Gynecology, Aretaieion Hospital, Medical School, National and Kapodistrian University of Athens, Vas. Sofias 76, 11528 Athens, Greece

**Keywords:** COVID-19, SARS-CoV-2, pregnancy, prevention, management, prenatal care, intrapartum care, postpartum care

## Abstract

Constant accumulation of data results in continuous updates of guidelines and recommendations on the proper management of pregnant women with COVID-19. This study aims to summarize the up-to-date information about the prevention and management of suspected/confirmed SARS-CoV-2 infection in obstetric patients and obstetric care during prenatal, intrapartum, and postpartum periods. We conducted a comprehensive literature search in PubMed for relevant English-written full-text reviews. We also included relevant guidelines and recommendations. In women with a low risk for infection and uncomplicated pregnancy, elective and non-urgent appointments should be postponed or completed through telehealth. Vaccination should be discussed and distance and personal hygiene preventive measures should be recommended. Routine ultrasound examinations should be adjusted in order to minimize exposure to the virus. Standardized criteria should evaluate the need for admission. Women with moderate/high-risk for infection should be isolated and tested with RT-PCR. The mode and timing of delivery should follow routine obstetric indications. In case of infection, glucocorticoids are recommended in critically ill pregnant women, after individualized evaluation. During labor and concomitant infection, the duration of the first two stages should be reduced as possible to decrease aerosolization, while minimization of hemorrhage is essential during the third stage. Close maternal monitoring and adequate oxygenation when necessary always remain a prerequisite. Discharge should be considered on the first or second day postpartum, also depending on delivery mode. Breastfeeding with protective equipment is recommended, as its benefits outweigh the risks of neonatal infection. Recommendations are currently based on limited available data. More original studies on infected pregnant women are needed to establish totally evidence-based protocols of care for these patients.

## 1. Introduction

Early in December 2019, viral pneumonia cases of unknown cause occurred in the city of Wuhan, the capital of Hubei province of central China. Soon enough, the outbreak was identified to be caused by a novel coronavirus labeled severe acute respiratory syndrome coronavirus 2 (SARS-CoV-2), and the clinical state was named as a new coronavirus disease 2019 (COVID-19). New confirmed cases outside China and associated mortality and morbidity rates have rapidly increased, showing that the disease was spreading to all continents and globally challenging health care systems. On 11 March 2020, the WHO declared COVID-19 as a pandemic infection [1]. By 6 April 2021, more than 131 million infections and 2.8 million deaths had been reported [2].

The current situation of the pandemic, with the daily rise of reported cases all over the globe, changed the routine of healthcare practices, as new challenges appeared; these affected obstetric and childbirth healthcare services, which cannot be suspended. As a new infection, evidenced-based knowledge about COVID-19 disease in pregnancy is limited, and the decisions about prevention, diagnosis, and management should be made based on previous experience with viral infections, clinical judgment, and common sense [3,4]. Anatomic, physiologic, and immunologic changes during pregnancy are considered to pose a risk for severe COVID-19 infection and disease, as was previously experienced with the H1N1 pandemic [5,6,7]. Increased maternal oxygen demand, elevated diagram, and edema of the respiratory mucosa during pregnancy decrease the maternal tolerance of hypoxia [8]. Therefore, pregnant women could be more likely to develop a more severe respiratory status (in the case of a viral pneumonia, which happened in the outbreak of H1N1, SARS-CoV-1, and middle east respiratory syndrome coronavirus (MERS-CoV)) [1,2,3,7,8,9]. As the course of this disease unfolds, national or international recommendations and guidelines for the prevention and management of SARS-CoV-2 infection in pregnancy are published by various scientific organizations and are constantly updated to correspond to the latest scientific data [10,11,12].

The aim of this review was to summarize the accumulated up-to-date recommendations on prevention and management (both pharmacological and non-pharmacological) of SARS-CoV-2 infection during pregnancy as well as obstetric care during prenatal, intrapartum, and postpartum periods.

## 2. Materials and Methods

We searched the PubMed Database for English-written, full-text narrative or systematic reviews on either pharmacological or non-pharmacological prenatal, intrapartum, and postpartum management of obstetric patients with suspected or laboratory-confirmed SARS-CoV-2 infection. The following keyword algorithm was used: (COVID-19 OR SARS-CoV-2) AND ((management AND pregnancy) OR “obstetric care” OR “prenatal care” OR “Antenatal care” OR (pregnancy AND treatment)). After the initial deduplication process, we conducted an initial title–abstract screening, excluding only totally irrelevant studies. During the second stage of full-text assessment, only studies fulfilling the aforementioned criteria were considered eligible for our review. Eligible articles were also identified through searching references of included studies, implementing the snowball procedure. Finally, we found it reasonable to check the guidelines of scientific perinatal committees, including the American College of Obstetricians and Gynecologists (ACOG: Washington, USA), International Federation of Gynaecology and Obstetrics (FIGO: London, UK), International Society for Ultrasound in Obstetrics and Gynecology (ISUOG: London, UK), Royal College of Obstetricians and Gynaecologists (RCOG: London, UK), Society of Maternal Fetal Medicine (SMFM: Washington, USA), and the United States Centers for Disease Control and Prevention (CDC: Atlanta, USA). All processes were undertaken by two sets of researchers, with disagreements being solved through consensus or the involvement of a senior researcher. As this is not a systematic review, quality assessment of the included studies was not performed.

## 3. Results

Figure 1 depicts the flow diagram of this review.

### 3.1. Initial Assessment and Management of Prenatal Period

Since several treatment methods remain under dispute, preventive measures are of utmost importance [13]. Vaccination at pregnancy should be considered with caution, as pregnant women have been excluded from initial major vaccination trials, so there is currently inadequate evidence assessing both its efficacy and safety [14]; thus, a robust recommendation cannot be given. In our view, new data are coming soon to fill this gap. Experts believe mRNA vaccines pose no risks to the pregnant woman or the fetus, mainly through previous knowledge of nonliving vaccines. Many international government agencies recommend COVID-19 vaccination during pregnancy or breastfeeding, especially in high-risk individuals who meet the relevant criteria [14,15]. The final decision should be made by pregnant women after being fully informed through adequate consultation from their attending clinicians about the potential benefit and unknown risks [15,16]. Since vaccination is not a routine practice in pregnancy and vaccines are not universally available, strict preventive measures should be followed. Particularly, the risk of transmission can be reduced by self-protection, regular distant communication of pregnant women with their clinician, patients triage regarding the risk level, and early quarantine of suspected and confirmed cases. Pregnant women should avoid crowded areas, such as means of transport, maintain a distance of almost two meters from other people, and restrict unnecessary travel. A significant component of self-protection is personal hygiene, for example, seven-step hand-washing [17,18]. Additionally, specific shielding measures with strict limitation of contact have been published by several communities, regarding especially vulnerable individuals for severe COVID-19 [19]. 

Relevant guidelines on prevention and management of pregnant women during the prenatal period have been developed by the Professional Perinatal Societies, such as ACOG [11], SMFM [20], ISUOG [3], RCOG [11], and the CDC [21] for care of pregnant women.

#### 3.1.1. Preadmission Screening

During the pandemic, screening and triage are essential in all outpatient and hospital care to prevent viral transmission. Antenatal care appointments should be precipitated by a screening process, through telephone calls. In case of the existence of clinical symptoms, the appointment should be delayed or a quantitative reverse-transcriptase polymerase chain reaction (qRT-PCR) test has to be recommended [22]. Testing should be performed according to laboratory protocols, eligibility, and the local epidemiological evaluation [3,23,24,25]. In cases of antenatal emergency episodes, or labor, patients should be transferred to a designated and isolated area for triaging, and a qRT-PCR test should be performed in case of admission [24]. Visits should be limited according to each hospital policy. Usually, only one support partner could be considered in the delivery ward after following the same screening procedure and having a negative qRT-PCR test [3,22,25]. For suspected or confirmed COVID-19 patients, supportive partners should not be permitted [4].

The screening process requires a specific area for triaging and an established algorithm to be followed by all health care workers (HCWs). The algorithm for case classification varies widely according to local case definitions; however, a similar rationale is followed in most cases [26]. Established questionnaires translated into different languages could be useful for the triage procedure [24]. The initial assessment of a pregnant woman who visits the hospital should start with the measurement of her temperature. If she has a fever or claims a former episode of high temperature, the presence of respiratory symptoms, such as cough or shortness of breath, should initially be investigated [24]. Otherwise, the patient is categorized as low-risk. If both fever and respiratory symptoms are not present, then data on epidemiological information, such as “travel history, occupation, significant contact, and cluster” (TOCC) risk factors, should be obtained [25]. The definition of “significant contact” varies between countries, as it is based on local protocols, yet a consensus on distance and duration limits might exist [3,26]. A positive response classifies the patient into the high-risk group and a negative one into the moderate-risk group. Pregnant women who are considered “low-risk” should receive the prearranged perinatal care or intrapartum care with the standard personal protective equipment (PPE) [23]. Guidelines for the moderate/high-risk group include isolation in a designated area and screening with a qRT-PCR test. Until results are acquired, all suspected patients should be treated as positive. HCWs should consider the possibility of a false-negative result after qRT-PCR testing, as most commercially available assays interpret the result with a qualitative method [26]. Additionally, there is a possibility of a false-positive result with a positive predictive value ranging from 47.3% to 96.8% and a negative predictive value from 96.8% to 99.9% [27]. The sensitivity could be increased with the combination of an IgM ELISA assay with a qRT-PCR test [28]. However, antibody tests alone are inferior in detecting the acute onset of infection [27,28]. If the result is negative, but clinical suspicion remains high, qRT-PCR should be repeated after 24 h [6,25,26]. Chest computed tomography (CT) should also be considered after an informed consent approval [3,25].

#### 3.1.2. Place of Care

All potential/confirmed COVID-19 women should be placed in single isolation rooms or negative pressure rooms, if available. There are certain initial criteria, which are followed, in order to evaluate whether a woman with COVID-19 needs admission or not. These criteria are fever of ≥38 °C, which persists even after paracetamol use, findings associated with pneumonia in chest X-ray, and CURB severity scale with a score of >0 (C: acute confusion, U: urea >19 mg/dL, R: ≥30 bpm, B: systolic blood pressure of ≤90 mm Hg or diastolic blood pressure of ≤60 mmHg) [3]. Pregnant women with other co-morbidities, such as chronic hypertension, immune system disorders, pulmonary diseases, use of immunosuppressive drugs, or diabetes are considered high-risk. Additional factors associated with ethnicity minorities and socioeconomic status may increase the disease’s burden and should therefore not be overseen [29,30]. Geographical differences might further affect the outcomes of COVID-19, due to differences in therapeutic management and adherence to preventive measures [31].

A pregnant woman needs admission to the intensive care unit if she needs vasopressors or mechanical ventilation [32,33]. In addition, the need for aggressive fluid resuscitation is established when three of the following criteria are met: respiratory rate of ≥30 bpm, PaO_2_/FiO_2_ ratio of <250, multilobar infiltrates, confusion, uremia (>20 mg/dL), leukopenia (<4000 cells/mm^3^), thrombocytopenia (<100,000 platelets/mm^3^), hypothermia/central temperature of <36 °C, and hypotension [33]. The decision regarding the woman’s admission is based on TOCC risk factors and clinical symptoms [32].

All pregnant women with a contact history with a COVID-19 positive person or temperature of ≥37.3 °C and respiratory symptoms should be admitted to an emergency internal medicine or specialist outpatient department fever clinic. If respiratory symptoms are present, admission to a respiratory department is recommended; if absent, women should be followed up according to temperature (≥37.3 and <37.3 °C, respectively) [17].

Generally, the majority of infected pregnant women with mild clinical presentations could be managed at home with symptomatic treatment, hydration, and rest [3]. Home isolation should be considered after assessing risk factors for disease severity and ensuring adequate monitoring of women’s symptoms [32]. Patients should be daily followed up for fever, respiratory rate, blood pressure, and fetal movements and keep a close tele-contact with their attending obstetrician [28]; in this context, personal monitoring equipment, such as pulse oximetry devices for home use, might be of help.

#### 3.1.3. Prenatal Appointments

Routine appointments for women with suspected/confirmed COVID-19 should be delayed until after the recommended period of self-isolation [22]. For symptomatic patients, postponement by 14 days if positive, or after two negative results, is advised. For patients with symptomatic household contacts appointments should be deferred for 14 days. Elective and non-urgent appointments should be postponed or completed by telehealth [25]. Although minimizing the unnecessary prenatal visits through scheduled virtual visits may reduce the risk of exposure to SARS-CoV-2, each system’s resources in terms of telehealth differ, rendering the global implementation of such a strategy extremely challenging [34].

##### Ultrasound Frequency and Equipment/Patient Room

If medically indicated and assuming that all protective measures (i.e., proper use of PPE by both the pregnant and physician, standardized disinfection of equipment per manufacturer guidelines after each use, and adequate ventilation of the examination room) are strictly followed, routine ultrasound scans (USS) should be performed, while unnecessary prenatal appointments should be avoided [6,25]. Deferment or cancellation of testing/examinations is advised if the risks of exposure and infection within the community outweigh their benefits. Completion of laboratory tests and inevitable USS should take place on the same visit day. Gestational dating and nuchal translucency USS should be combined in the same visit during the first trimester. In the second trimester, between 20–22 weeks, a serum triple screen (alpha-fetoprotein, human chorionic gonadotropin, and unconjugated estriol) and an anatomy scan are recommended to be performed in one visit [17]. Serial cervical length after anatomy USS in cases of transvaginal USS cervical length above 35 mm and previous preterm birth at >34 weeks is not necessary. Single growth follow-up at 32 weeks and low-lying placenta follow-up at 34–36 weeks is recommended [6,25]. In the case of a positive patient, both the room and instruments need to be disinfected cautiously and according to standardized procedures [35].

##### GBS Screening

Group B streptococcus (GBS) screening is indicated at 36 weeks of gestation. It can be delayed by 14 days in patients with TOCC risk factors [32].

##### Antenatal Surveillance (Biophysical Profile, Non-Stress Test)

In high-risk pregnancies, such as those with fetal growth restriction with abnormal umbilical arterial Doppler studies, complicated monochorionic twins, or Kell-sensitized patients with significant titers, a twice-weekly non-stress-test (NST) is suggested. A biophysical profile should be performed instead of NST if a patient needs USS. Daily NST is suggested only for hospitalized patients [25].

### 3.2. Labor and Delivery

#### 3.2.1. Pre-Delivery Preparation

In spite of the pandemic, labor admissions cannot be suspended and are anticipated at any time in all the hospitals with obstetric care units. To limit viral exposure, women should be instructed to stay away from work for two weeks before the anticipated date of delivery (at about 38 weeks) and practice strict social distancing [36]. Nevertheless, such a measure may be accompanied by certain deficits, when for example during the home “isolation”, pregnant women inattentively interact with their family members. Practical suggestions to prevent the potential of viral transmission by a family member include the wearing of face masks, regular disinfection of high contact surfaces with >60% ethanol, regular handwashing, and use of disposable–personal items [27]. These measures should be applied by both pregnant women and their household partners [28].

When admitted to the labor ward, all women should undergo risk stratification for potential COVID-19 [26]. Women with positive history or with any associated symptoms are considered as “person under investigation” cases and should be managed accordingly [23]. In case of scheduled cesarean delivery or induction of labor, the screening process should be performed by phone before admission, to reduce the possibility of viral transmission to care units [37]. Nevertheless, HCWs ought to take into account the possibility of false-negative test results [26]. The screening process may vary between hospitals depending on local protocols, prevalence of infection, testing availability, and laboratory response time [33]. The universal testing approach with qRT-PCR test prior to labor admission is considered reasonable, due to the high rate of asymptomatic patients who can spread the virus [28,38]. The test should be performed 24–48 h before hospitalization so that the results will be available by the time of admission [24]. In the case of an urgent condition, screening should be performed in an admission ward right away, and management should not be delayed while waiting for the results [25]. Due to the possibility of viral transmission, the suspected pregnant women should be treated as positive [26]. Pregnant women are restricted to only one consistent asymptomatic adult partner at the time of labor and delivery, always wearing a mask and undergoing the same screening procedure [36]. In terms of necessary infrastructure, designated isolation rooms and operating rooms should be established, with negative pressure or with a filtration system such as the high-efficiency particulate absorbing units if possible [4]. Moreover, unnecessary transportation of the patient in the hospital should be avoided, and dedicated corridors from labor rooms to maternity units or operating rooms should be considered [39]. Alternatively, the use of convertible rooms could avoid transfer throughout the whole delivery procedure [33]. The availability of all the aforementioned facilities in developing countries is limited, due to low economic resources for healthcare provision, which predisposes obstetric patients with COVID-19 to an increased risk for adverse outcomes [30].

HCWs should be notified of the suspected or confirmed COVID-19 cases and take all appropriate precautions to reduce the risk of infection. All involved HCWs should follow the regularly updated CDC and WHO recommendations for hand hygiene and use of PPE (i.e., medical mask, protective eyewear, gloves, and gown) [4]. Nonetheless, their global execution seems, once again, really challenging, as reflected by the differences in the availability of equipment and resources between countries [28].

Viral transmission has been proven to occur via surface fomites and droplet infection, especially after prolonged close contact [28,38]. In contrast, limited evidence suggests clear airborne transmission during aerosol-generating procedures. but the possibility of exposure should not be ruled out [28]. Therefore, and in the case of a suspected or confirmed COVID-19 case, N95 masks or FFP2 respirators are necessary to use during aerosolization procedures, such as general anesthesia, forceful pushing of the second stage of labor, and oxygen supplementation [25]. Some guidelines suggest using N95/FFP2 respirators always in case of close contact with a suspected or confirmed case and regardless of the potential for aerosolization [28].

In order to develop multidisciplinary management and limit the exposure of health care personnel, all institutions should run constantly updated stimulations for elective or emergency scenarios [40]. The simulations should be conducted on a regular basis in hospitals with the minimum time and number of staff required and include correct donning, doffing, and disposal of PPE [39].

#### 3.2.2. Time and Mode of Delivery

At the onset of the pandemic, cesarean section (CS) was the delivery mode of choice in infected pregnant women and frequently before term gestation [41]. The possible explanation for this decision was to reduce the possible risks of unstable maternal respiratory status, fetal demise, and viral transmission [6,42]. However, the role of CS in reducing these risks has not been established [42,43]. Therefore, COVID-19 alone is not an indication for pregnancy termination or CS. The time and mode of delivery should only be based on routine obstetric indications [25]. Critically ill patients require close monitoring of maternal and fetal status, and the decision regarding delivery should be made by a multidisciplinary team depending on maternal and fetal benefits [12,44]. Early controlled delivery should be considered when maternal and/or fetal well-being deteriorate despite optimal supportive care [44]. Nevertheless, it is unclear whether delivery in such cases improves the maternal respiratory status unless maternal and fetal benefits outweigh the operative risks and complications of prematurity [45].

As long as a suspected or confirmed COVID-19 case is asymptomatic or has mild symptoms and a stable respiratory status, induction of labor and scheduled CS should not be postponed when there is appropriate obstetric indication [36]. This may not be possible when there is no bed available. When a patient is infected in her late pregnancy, scheduled deliveries should be postponed until she recovers or a negative testing result is obtained. Pregnant women in active labor cannot be delayed, although should be encouraged to stay home during the early stage [25].

#### 3.2.3. Intrapartum Management

In suspected or confirmed cases of COVID-19, intrapartum management requires reform in order to assure fetal and maternal benefit along with minimal risks for the healthcare providers. Under normal labor progression, the first stage mainly follows the same standard practice, and reduction of the duration of exposure is quite important. Cervical examination and the USS assessments should be reduced as much as possible [4]. As mentioned above, CS should not be expedited in case of labor arrest but should be performed according to standard obstetric protocols [36].

The second stage of labor is regarded as an “aerosolization procedure”, as the pushing process increases the patient’s respiratory secretions. Therefore, the second stage should be shortened, and pushing motions should not be delayed and should accompany uterine contractions [33]. When prerequisites are met, an operative vaginal delivery should be considered. In this way, the respiratory effort of the mother is reduced, and the possible hypoxic condition after exhaustion is prevented [4,25]. During the third stage, reduction of postpartum hemorrhage should be prevented or immediately managed, as, due to the pandemic, there is restriction of donated blood supplies [6].

Current evidence indicates a very small risk of intrauterine vertical transmission or transmission during delivery [28]. However, as the data remain limited, the probability of vertical transmission cannot be ruled out as yet [41]. Certain precautions should be taken, as the newborns may also acquire the infection after delivery [27,38]. It is prudent to avoid fetal scalp pH examination, internal fetal heart rate monitoring, and routine umbilical cord gas analysis [33]. Moreover, avoidance of delayed umbilical cord clamping, when there is no contraindication, is suggested by several prenatal society guidelines [25]. In contrast, RCOG guidelines [10] state that practicing early cord clamping cannot decrease the already existing possibility of transmission, so the delayed cord clamping should be performed as a standard of care [6,46]. After delivery, any material, especially a biological sample such as the placenta, should be regarded as contaminated and should be treated accordingly [33,42,47].

SARS-CoV-2 infection is not a contraindication for regional or for general anesthesia. Regional anesthesia is actually encouraged, as it reduces maternal respiratory stress due to pain and anxiety during labor. It should even be provided early in order to avoid general anesthesia, and the subsequent aerosol-generating procedure of intubation, in the case of an emergency CS [33]. In order to reduce the possibility of anesthetic conversion, de novo spinal anesthesia or combined spinal and epidural anesthesia may be utilized. When general anesthesia is necessary, it should be performed by the most experienced staff and with the assistance of videolaryngoscopy to achieve first-pass success and minimize the exposure [47,48]. Analgesia with inhalation of nitrous oxide is not clearly recognized as an aerosolization procedure, and most of the equipment’s circuits contain antiviral filters. Nevertheless, its extensive use is discouraged by some scientific societies, due to the potential risk of viral transmission [25]. Moreover, intravenous opioid analgesia could lead to the deterioration of maternal respiratory status; therefore, its use in labor should be suspended. At the same time, maternal vital signs, fluid input and output, and NST should be continuously monitored [33,45].

Oxygen supplementation is a controversial issue, as it combines a non-generally accepted fetal benefit (i.e., intrauterine resuscitation) with an increase in maternal respiratory droplets [4]. Therefore, oxygen masks should be used cautiously and in accordance with maternal oxygen saturation and fetal heart rate tracing [45]. Anticoagulation prophylaxis with heparin infusion should be considered in severely infected pregnant women during delivery, and also in cases of venous thromboembolism and pulmonary embolism in which the respiratory status of the patient is expected to deteriorate dramatically [22]. Antenatal corticosteroids are still used as usual in pregnant women who meet the criteria for enhancement of fetal lung maturation [7]. Their administration should be decided on an individualized basis. In late preterm (>34 weeks) gestation, corticosteroids exhibit no clear fetal benefit and should be avoided [4,23]. Sulfate magnesium, which is indicated for neuroprotection of premature (<32 weeks) fetuses, has a respiratory depressant side effect that is potentially harmful to a confirmed COVID-19 case. The SMFM has recommended alternative dosing in cases of already mild respiratory distress [45].

#### 3.2.4. Postpartum Management

After delivery, early discharge and continuation of isolation are encouraged. In uncomplicated cases, discharge is recommended during the first day after vaginal delivery and the second day after CS. Postpartum pain control in COVID-19 confirmed cases is still managed with nonsteroidal anti-inflammatory drugs, as there is no evidence of them worsening the patients’ condition. Postpartum visits and psychological support may be provided through a telehealth service [4,24,25]. Virtual visits have achieved comparable health outcomes with live visits, on the condition that they follow a multifaceted approach [37].

The neonate should be dried, cleaned, disinfected, and taken to an isolation room, with a swamp sample being already acquired for qRT-PCR testing [8]. The absence of mother-to-neonate transmission evidence has generated controversies with regard to maternal and neonatal separation. Skin-to-skin contact is mainly encouraged by prenatal societies guidelines when precautions are taken but should be reconsidered when maternal status is critical. The CDC suggests a healthy caregiver if possible for caring for the newborn when the mother is in isolation due to COVID-19 [21]. In all cases, the mother should be fully informed by her HCWs of the risks and benefits, and she should make the final decision [6]. Finally, current evidence supporting viral transmission through the breast milk of infected mothers is limited [7]. Therefore, natural breastfeeding is still recommended, as its benefits outweigh the risks, with hand washing, wearing a face mask, and avoiding coughing when handling the baby remaining prerequisites [6]. If the mother wishes for the separation from her newborn, she is encouraged to use a dedicated breast pump. Instructions should, once again, be given for proper hand hygiene and disinfection of the pump after each use [7].

All recommendations on obstetric care discussed above are summarized in Table 1.

### 3.3. Pharmacological Management of Infection in Pregnancy

Several pharmacological agents have been evaluated for their effectiveness against COVID-19 infection since the pandemic outbreak. Importantly, pregnant women have been mainly excluded from most of the clinical trials; thus, pharmacological maternal treatment is challenging, as there are minimal safety data available [49].

Lopinavir–ritonavir and chloroquine/hydroxychloroquine were considered as candidate options for treatment of SARS-CoV-2 infection and could be applied in pregnancy as well, as they have not been associated with significant adverse neonatal outcomes [6,26,50,51]. However, new available data indicate doubt that there is a clear therapeutic benefit from their use in COVID-19 patients or support that the benefit is outweighed by their toxicity [52,53].

Remdesivir is an antiviral agent, being initially proven to control in vitro viral replication and therefore forming the rationale to be tested for clinical use [54]. On 1 May 2020, the drug was approved for hospitalized children ≥12 years and adults with COVID-19 [33,51]. However, there is insufficient evidence for its safety during pregnancy, mainly due to its unknown risk of transplacental transfer [55,56]. Most relevant trials have excluded pregnant and breastfeeding women, although high recovery rates were observed in pregnant women who received compassionate use of remdesivir [57]. Therefore, standard-of-care administration of remdesivir in pregnant women with COVID-19 should be currently avoided [42,56].

Tocilizumab is an interleukin-6 receptor antagonist, which may counteract the inflammatory response in patients with severe COVID-19. Studies have demonstrated that tocilizumab is safe and efficacious in reducing mortality among critically ill patients [58,59]. However, due to the limited number of observational studies with considerable heterogeneity, its effectiveness needs to be further investigated through adequately conducted randomized clinical trials [60]. There is also currently no compelling evidence that it is linked to fetal malformations, but further investigation is required to confirm the benefit–risk and safety profile of tocilizumab in pregnant women [6,9,50,55].

The use of glucocorticoids was initially discouraged, as it was associated with unfavorable outcomes in COVID-19 positive patients [61]. Until recently, WHO recommended against the routine use of systematic corticosteroids, as it appeared that they delay viral clearance without important survival benefits [26,50]. In contrast, new evidence supported that dexamethasone can provide significant reduction in mortality for individuals with COVID-19 requiring mechanical ventilation or supplemental oxygen [62]; thus, pregnant women meeting these criteria could be considered for corticosteroid use. Unfortunately, dexamethasone has exhibited a higher rate of placental transfer than other glucocorticoids, so that its extended administration may lead to adverse neonatal effects. In contrast, other glucocorticoids, such as methylprednisolone and hydrocortisone, have resulted in less fetal steroid exposure, so that they could be administered as an alternative option for maternal treatment [63]. According to the RCOG guidelines, oral prednisolone or intravenous hydrocortisone for 10 days or until hospital discharge can be used. If steroids are indicated for the enhancement of fetal lung maturity, dexamethasone is recommended in the usual doses and then replaced with oral prednisolone or hydrocortisone to complete a total of 10 days in the cases of severe maternal disease [10].

Venous thromboembolism prophylaxis should be considered in hospitalized patients with COVID-19 due to the associated risk of excessive inflammation, hypoxia, immobilization, and diffuse intravascular coagulation [4,22]. As pregnancy per se is linked to a hypercoagulable state, obstetric patients that are admitted to hospital should be given anticoagulation treatment [64]. Low-molecular-weight heparin is recommended in prophylactic doses after an individual-based decision according to relevant risk factors and meticulous evaluation of possible contraindications [22].

Administration of antibiotics, such as amoxicillin, azithromycin, and ceftriaxone, which are commonly used in pregnancy, is recommended in cases in which there is a clinical suspicion of coexistent bacterial infection or sepsis [50].

Supportive therapy is an important part of the management of every infection with balanced nutrition, hydration, adequate levels of electrolytes, and rest [32]. According to the severity of infection, supplement oxygen inhalation in 60–100% concentration is recommended to be given at a rate of 40L/min. It is also necessary to check vital signs and oxygenation closely so that there is the ability to promptly react to any disorder or imbalance [18].

Figure 2 summarizes the main COVID-19-related recommendations from the time that a pregnant woman arrives in the hospital until she leaves the delivery room.

## 4. Discussion

This is a narrative summary of the accumulated up-to-date recommendations on the initial assessment and management of the prenatal period and the pharmacological and non-pharmacological management of infected pregnant women according to treatment options and the available obstetrical management strategies during childbirth. We have synthesized relevant full-text narrative and systematic reviews on these issues, adding data from the currently published guidelines of scientific perinatal committees.

The prevention, which includes personal hygiene, social distancing, and early quarantine of confirmed or suspected cases, is of absolute necessity. As part of preventive measures, vaccination should always constitute a fundamental part of the advice given to pregnant women and should be accompanied by proper guidance, always in accordance with the updated literature [14]. Screening and triage, based on measurement of temperature, respiratory symptoms, and TOCC, are necessary in every hospital [23,24]. Telehealth along with recommendations for fewer unnecessary visits during pregnancy aim at minimizing face-to-face contact [37]. Negative pressure or single isolation rooms ought to be available [4]. Clinical state and risk factors should always be accounted for to determine the necessity of admission and close monitoring [3,24].

In case of hospitalization, supportive care with close monitoring of vital signs and respiratory status is required [44]. Methylprednisolone and hydrocortisone should be considered in critically ill pregnant women [63]. Further research about the safety and effectiveness of remdesivir, tocilizumab, and glucocorticoids during pregnancy is mandatory [49]. Home isolation two weeks before the anticipated date of labor along with SARS-CoV-2 qRTPCR testing 24–48 h prior to admission are suggested [24]. COVID-19 is not an official indication for CS [25]. During labor, good maternal oxygenation, close monitoring of the fetus, and minimization of exposure to the staff are essential [36]. Health care professionals ought to wear PPE when in contact with the patient as well as N95 masks or an FFP2 respirator during aerosolization procedures [4].

There are not sufficient data to rule out the possibility of vertical transmission, so precautions during delivery are recommended. Breastfeeding is encouraged, as its benefit outweighs the theoretical risk of viral transmission to the neonate through contact [7]. Evolving evidence on COVID-19 and its impact on all trimesters of pregnancy will solidify more evidence-based management protocols in the future so that healthcare providers and pregnant women can navigate through this unprecedented situation.

### Strengths and Limitations

The main strength of this review is that it attempts to summarize currently available evidence and recommendations on the management of obstetric patients, covering the entire preventive, diagnostic, and therapeutic spectrum during prenatal, intrapartum, and postpartum periods. It further combines available literature with formal guidelines published by eminent scientific societies. Our study is of course subject to certain limitations, primarily associated with its non-systematic nature. However, the narrative flow was inevitable when considering the heterogeneity of aspects that this review aimed at addressing. When reviewing current recommendations, experts’ opinions may be included, despite the risk of low-quality data. Finally, the gaps in current knowledge and the constantly changing evidence require regular assessment and updating of the current guidelines. Therefore, the information presented in this review is not absolute and will be modified over time.

## 5. Conclusions

The lack of solid data on the therapeutic management of COVID-19 in pregnancy underlines the value of preventive measures. All current guidelines and recommendations on obstetric care follow that axis, focusing on the minimization of pregnant women’s exposure to healthcare settings, the existence of infection screening protocols and proper isolation infrastructures, careful surveillance of infected patients, use of personal protective equipment during all visits, and expedition of obstetric procedures during all prenatal, labor and delivery, and intrapartum periods. Further original studies on the management of infected pregnant women are warranted in order to establish more solidified evidence-based universal guidelines.

## Figures and Tables

**Figure 1 healthcare-09-00467-f001:**
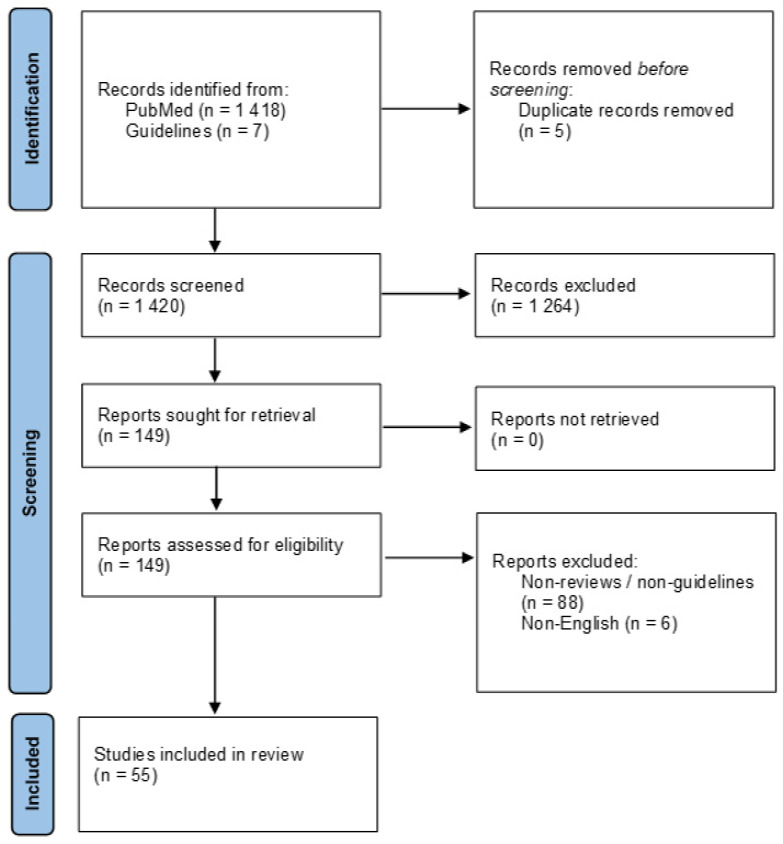
Flow diagram of this review.

**Figure 2 healthcare-09-00467-f002:**
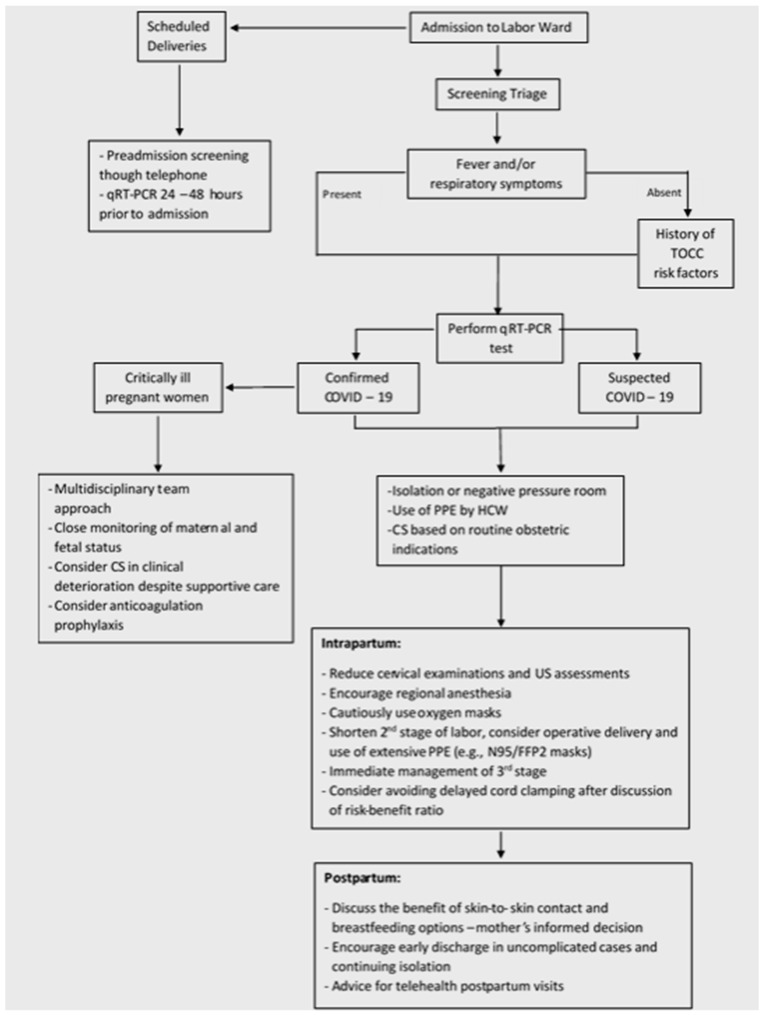
An illustrative summary of COVID-19-related protocols on management of pregnant women during the pandemic. The main recommended actions from admission to discharge and immediate postpartum period are depicted. Abbreviations: COVID-19: Coronavirus Disease 2019; TOCC: travel, occupation, significant contact, and cluster; qRT-PCR: quantitative reverse transcriptase–polymerase chain reaction; PPE: personal protective equipment; HCWs: health care workers; US: ultrasound; CS: cesarean section.

**Table 1 healthcare-09-00467-t001:** Recommendations for obstetric care during the COVID-19 pandemic.

Prenatal Care	Inform about preventive measures and inform about vaccination. Use screening triage, assess for respiratory symptoms and TOCC risk factors, and perform qRT-PCR test. Evaluate admission criteria, isolate in negative room suspected/confirmed cases, and provide supportive therapy (fluid and electrolyte balance, oxygen supplementation). Encourage prenatal visits through telehealth and limit face-to-face appointments. Combine dating and nuchal translucency US in the first trimester and a serum triple screen and an anatomy scan in the second trimester. Consider grouping GBS screening with other visits at 36 weeks of gestation. Perform twice weekly NST only in high-risk pregnancies (fetal growth restriction with abnormal umbilical arterial Doppler studies, complicated monochorionic twins, or Kell-sensitized patients with significant titers).
Intrapartum Care	Advise strict social distancing two weeks prior to anticipated day of delivery. Provide negative pressure isolation rooms in suspected or confirmed cases and use PPE. Decide on delivery mode and time according to routine obstetric indications. Use N95 masks or FFP2 respirators during aerosolization procedures. Closely monitor maternal and fetal status in critically ill pregnant women. Minimize intervals of cervical exams and US assessments. Consider early administration of regional anesthesia in suspected or confirmed cases. Consider shortening the second stage of labor. Control postpartum bleeding by active management of third stage. (blood shortage).
Postpartum Care	Discharge first day after VD/second day after CS. Encourage postpartum visits and phycological support through telehealth service. Encourage mother to undertake skin-to-skin contact and breastfeeding. Discuss risks and benefits. Advise mother to practice hand and tissue hygiene, avoid coughing, and wear a mask while feeding. Advise breast pumping during separation.

Abbreviations: COVID-19, Coronavirus Disease-2019; TOCC: travel history, occupation, significant contact, and cluster; qRT-PCR: quantitative reverse transcriptase–polymerase chain reaction; GBS: Group B Streptococcus; PPE: personal protective equipment; HCWs: health care workers; US: ultrasound; VD: vaginal delivery; CS: cesarean section.

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
