# Peer review of "Management and Prevention of COVID-19 in Pregnancy and Pandemic Obstetric Care: A Review of Current Practices"

_healthcare, 2021, doi:10.3390/healthcare9040467_

Round 1
Reviewer 1 Report
I read with great interest the review ‘Management and prevention of COVID-19 in pregnancy and pandemic obstetric care: a review of current practices’. The authors aimed to to summarize currently available evidence and recommendations on management of obstetric patients, covering the entire preventive, diagnostic and therapeutic spectrum during prenatal, intrapartum and post-partum periods.
Obviously, the topic is of great relevance and interest. The paper is well-written in general, well structured and easy to read.
I have some concerns regarding a number of points. Firstly, the vaccination recommendation is one-way and leaves no room for different opinions (‘Vaccination should be recommended by obstetricians in all women being pregnant, planning to become pregnant, breastfeeding or planning to breastfeed, and always with unequivocal scientific justification and with careful counseling in cases of vaccine hesitancy’. )
On the other hands the authors are more cautious about other recommendations. E.g. ’ …avoidance of delayed umbilical cord clamping, when there is no contraindication, is suggested by several prenatal society guidelines [22]. In contrast, RCOG guidelines, 10 state that practicing early cord clamping cannot decrease the already existing possibility of…’; ‘…Pharmacological management of infection in pregnancy. .. pregnant women have been mainly excluded from most of the clinical trials, thus pharmacological maternal treatment is challenging, as there is little safety data available’.
In paragraph 3.1.3.1. (Ultrasound frequency and equipment / patient room ), forth line, probably in is ‘ the same visit’, not in the ‘dame visit’
In conclusion we recognize the value of the paper, we only suggest the author to better describe our knowledge on vaccination for COVID-19 during pregnancy.
Author Response
We would like to thank the reviewer for taking the time and effort to assess our original submission so meticulously. We have taken into account all of your comments and recommendations and we have modified our paper accordingly. All manuscript changes have been highlighted using the “tracked changes” function provided by Microsoft Word. Detailed replies to the reviewers’ comments are provided below:
Point 1: “I have some concerns regarding a number of points. Firstly, the vaccination recommendation is one-way and leaves no room for different opinions”
Authors’ reply: Thank you for a valuable recommendation that will definitely improve our work. We added a new paragraph in page 3, section 3.1, line 97-105, where we elaborate more about current evidence on vaccination for COVID-19 during pregnancy. This new paragraph reads as follows:
“Vaccination at pregnancy should be currently considered with caution, as pregnant women have been excluded in major vaccinations trials, so that there is inadequate evidence assessing both its efficacy and safety [14]: thus, a robust recommendation cannot be given during pregnancy. In our view, new data are coming soon to fill this gap. Experts believe that mRNA vaccines pose no risks to both pregnant women and fetuses, mainly through previous knowledge on nonliving vaccines [14,15]. Many international government agencies recommend COVID-19 vaccination during pregnancy or breastfeeding especially in high-risk individuals who meet the relevant criteria. The final decision should be made by pregnant women after being fully informed through adequate consultation from their clinicians about potential benefits and unknown risks [15,16]. Since vaccination is not a routine practice in pregnancy and vaccines are not universally available, strict preventive measures should be followed. Particularly, the risk of transmission can be reduced by self-protection, regular distant communication of pregnant women with their clinician, patients triage regarding the risk level, and early quarantine of suspected and confirmed cases.”
Point 2: In paragraph 3.1.3.1. (Ultrasound frequency and equipment / patient room), forth line, probably in is ‘the same visit’, not in the ‘dame visit’
Authors’ reply: Thank you for dedicating time and effort to meticulously review our work. We have replaced the ‘dame visit’ with ‘same visit’ on page 5, paragraph 3.1.3.1, line 212.
We would like to inform the reviewer that we have added 12 new references in order to address reviewer’s recommendations and further incorporate current data. The aforementioned references correspond to the numbers 15, 16, 17, 19, 27, 28, 29, 30, 31, 34, 35, 38 and the rest of numbers has been modified accordingly. The new references are the following:
[15] Joint Committee on Vaccination and Immunisation., Advice on priority groups for COVID-19 vaccination. 2020.
[16] Rasmussen, S.; Denise, J., Pregnancy, postpartum care, and Covid-19 vaccination in 2021. Infect Dis 2020, 26(11), 2787-2789.
[17] Chen, Y.; Li, Z.; Zhang, Y. Y.; Zhao, W. H.; Yu, Z.Y., Maternal health care management during the outbreak of coronavirus disease 2019. Journal of medical virology 2020, 92(7), 731-739.
[19] Schmidt-Sane, M.; Jones, L.; Tulloch, O., Key Considerations: Emerging Evidence on Shielding Vulnerable Groups During COVID-19. 2020.
[27] Wu, D.; Fang, D.; Wang, R.; Deng, D.; Liao, S., Management of Pregnancy during the COVID‐19 Pandemic. Global Challenges 2021, 5(2), 2000052.
[28] Donders, F.; Lonnée-Hoffmann, R.; Tsiakalos, A.; Mendling, W.; Martinez de Oliveira, J.; Judlin, P.; Xue, F.; Donders, G. GG.; Isidog Covid-Guideline Workgroup., ISIDOG recommendations concerning COVID-19 and pregnancy. Diagnostics 2020 10(4), 243.
[29] Joseph, N. T.; Stanhope, K. K.; Badell, M. L.; Horton, J. P.; Boulet, S. L.; Jamieson, D. J., Sociodemographic Predictors of SARS-CoV-2 Infection in Obstetric Patients, Georgia, USA. Emerging infectious diseases 2020, 26(11), 2786.
[30] Amorim, M. M. R.; Takemoto, M. L. S.; Fonseca, E. B., Maternal deaths with coronavirus disease 2019: a different outcome from low-to middle-resource countries? American Journal of Obstetrics & Gynecology 2020, 223(2), 298-299.
[31] Dubey, P.; Thakur, B.; Reddy, S.; Martinez, C. A.; Nurunnabi, M.; Manuel, S. L.; Chedda, S; Bracamontes, C.; Dwivedi, A.K., Current trends and geographical differences in therapeutic profile and outcomes of COVID-19 among pregnant women-a systematic review and meta-analysis. BMC pregnancy and childbirth 2021, 21(1), 1-16.
[34] Fryer, K.; Delgado, A.; Foti, T.; Reid, C. N.; Marshall, J., Implementation of obstetric telehealth during COVID-19 and be-yond. Maternal and child health journal 2020, 24(9), 1104-1110.
[35] Bourne, T.; Kyriacou, C.; Coomarasamy, A.; Al‐Memar, M.; Leonardi, M.; Kirk, E.; Landolfo C.; Blanchette-Porter, M.; Small, R.; Condous, G.; Timmerman, D., ISUOG Consensus Statement on rationalization of early‐pregnancy care and provision of ul-trasonography in context of SARS‐CoV‐2. Ultrasound in Obstetrics & Gynecology 2020.
[38] Ashokka, B.; Loh, M. H.; Tan, C. H.; Su, L. L.; Young, B. E.; Lye, D. C.; Biswas, A.; Illanes, S. E.; Choolani, M., Care of the pregnant woman with coronavirus disease 2019 in labor and delivery: anesthesia, emergency cesarean delivery, differential diagnosis in the acutely ill parturient, care of the newborn, and protection of the healthcare personnel. American journal of obstetrics and gynecology 2020, 223(1), 66-74.
Reviewer 2 Report
Pountoukidou et al article reviews the current evidence on the management of pregnancy and labor during the COVID-19 pandemic. It draws out some of the most relevant recommendations for consideration in obstetric units. I think it is a great opportunity to learn what is in the scientific evidence to better adjust to the treatment and care of pregnancy. I guess it is a great effort made by the authors and it is very well written. I would simply note a few suggestions:
- Although it is not a systematic review, the flow-chart followed in the search and selection of the articles, as well as how many were extracted at each stage and what final "n" the authors analyzed, could appear in the material and methods section.
- There are some type errors such as the phrase "Error! Reference source not found" appearing multiple times, the acronym covid-19 appears in lower case (page 8) and the acronym CT on page 3 is not defined.
- Table 1 is very informative. I would suggest not to center the columns but to left justify them. Would the authors dare to make a figure with the recommendations from the time a pregnant woman arrives to the hospital until she leaves the delivery room, taking into account the COVID-19 protocol?
Author Response
We would like to thank the reviewer for taking the time and effort to assess our original submission so meticulously. We have taken into account all of your comments and recommendations and we have modified our paper accordingly. All manuscript changes have been highlighted using the “tracked changes” function provided by Microsoft Word. Detailed replies to the reviewers’ comments are provided below:
Point 1: “Although it is not a systematic review, the flow-chart followed in the search and selection of the articles, as well as how many were extracted at each stage and what final "n" the authors analyzed, could appear in the material and methods section”
Authors’ reply: Thank you for your insightful recommendation. Figure 1, that depicts the flow diagram of this review, has been added on page 3, lines 88-93. We found proper to update our search, so that it corresponds to the most recent literature (last search date: April 2, 2021).
Point 2: “There are some type errors such as the phrase "Error! Reference source not found" appearing multiple times”
Authors’ reply: Thank you for your kind comment. We have edited the text and have removed any type error of this kind.
Point 3: “the acronym covid-19 appears in lower case (page 8) and the acronym CT on page 3 is not defined.”
Authors’ reply: Thank you for your observation. The acronym covid-19 has been replaced with uppercase on page 10, line 398 and the acronym CT has been defined as ‘Computed Tomography” on page 4, line 158.
Additionally, all abbreviations in the text have been checked.
Point 4: “Table 1 is very informative. I would suggest not to center the columns but to left justify them.”
Authors’ reply: Thank you for your suggestion. We left justified the columns on Table 1, on page 9, lines 392-395.
Point 5: “Would the authors dare to make a figure with the recommendations from the time a pregnant woman arrives to the hospital until she leaves the delivery room, taking into account the COVID-19 protocol?”
Authors’ reply: Thank you for dedicating time and effort to meticulously review our manuscript. We have formulated Figure 2, that summarizes the main COVID-19 – related recommendations from the time a pregnant woman arrives to the hospital until she leaves the delivery room. We have inserted it on page 11, lines 456-463.
We would like to inform the reviewer that we have added 12 new references in order to address reviewer’s recommendations and further incorporate current data. The aforementioned references correspond to the numbers 15, 16, 17, 19, 27, 28, 29, 30, 31, 34, 35, 38 and the rest of numbers has been modified accordingly. The new references are the following:
[15] Joint Committee on Vaccination and Immunisation., Advice on priority groups for COVID-19 vaccination. 2020.
[16] Rasmussen, S.; Denise, J., Pregnancy, postpartum care, and Covid-19 vaccination in 2021. Infect Dis 2020, 26(11), 2787-2789.
[17] Chen, Y.; Li, Z.; Zhang, Y. Y.; Zhao, W. H.; Yu, Z.Y., Maternal health care management during the outbreak of coronavirus disease 2019. Journal of medical virology 2020, 92(7), 731-739.
[19] Schmidt-Sane, M.; Jones, L.; Tulloch, O., Key Considerations: Emerging Evidence on Shielding Vulnerable Groups During COVID-19. 2020.
[27] Wu, D.; Fang, D.; Wang, R.; Deng, D.; Liao, S., Management of Pregnancy during the COVID‐19 Pandemic. Global Challenges 2021, 5(2), 2000052.
[28] Donders, F.; Lonnée-Hoffmann, R.; Tsiakalos, A.; Mendling, W.; Martinez de Oliveira, J.; Judlin, P.; Xue, F.; Donders, G. GG.; Isidog Covid-Guideline Workgroup., ISIDOG recommendations concerning COVID-19 and pregnancy. Diagnostics 2020 10(4), 243.
[29] Joseph, N. T.; Stanhope, K. K.; Badell, M. L.; Horton, J. P.; Boulet, S. L.; Jamieson, D. J., Sociodemographic Predictors of SARS-CoV-2 Infection in Obstetric Patients, Georgia, USA. Emerging infectious diseases 2020, 26(11), 2786.
[30] Amorim, M. M. R.; Takemoto, M. L. S.; Fonseca, E. B., Maternal deaths with coronavirus disease 2019: a different outcome from low-to middle-resource countries? American Journal of Obstetrics & Gynecology 2020, 223(2), 298-299.
[31] Dubey, P.; Thakur, B.; Reddy, S.; Martinez, C. A.; Nurunnabi, M.; Manuel, S. L.; Chedda, S; Bracamontes, C.; Dwivedi, A.K., Current trends and geographical differences in therapeutic profile and outcomes of COVID-19 among pregnant women-a systematic review and meta-analysis. BMC pregnancy and childbirth 2021, 21(1), 1-16.
[34] Fryer, K.; Delgado, A.; Foti, T.; Reid, C. N.; Marshall, J., Implementation of obstetric telehealth during COVID-19 and be-yond. Maternal and child health journal 2020, 24(9), 1104-1110.
[35] Bourne, T.; Kyriacou, C.; Coomarasamy, A.; Al‐Memar, M.; Leonardi, M.; Kirk, E.; Landolfo C.; Blanchette-Porter, M.; Small, R.; Condous, G.; Timmerman, D., ISUOG Consensus Statement on rationalization of early‐pregnancy care and provision of ul-trasonography in context of SARS‐CoV‐2. Ultrasound in Obstetrics & Gynecology 2020.
[38] Ashokka, B.; Loh, M. H.; Tan, C. H.; Su, L. L.; Young, B. E.; Lye, D. C.; Biswas, A.; Illanes, S. E.; Choolani, M., Care of the pregnant woman with coronavirus disease 2019 in labor and delivery: anesthesia, emergency cesarean delivery, differential diagnosis in the acutely ill parturient, care of the newborn, and protection of the healthcare personnel. American journal of obstetrics and gynecology 2020, 223(1), 66-74.
Reviewer 3 Report
This is a major undertaking because of the wide scope of the article and its attempt to cover all aspects of pregnancy. The difficulty this generated is the lack of critical analysis of the often contradictory guidance or guidance that is not based on a sufficiently robust data base. An additional problem is that it does not take into consideration that the practice of maternity care varies widely across the globe. Indeed it varies widely within countries. So the applicability of the many of the elements explored will remain limited.
Some sentences need revision e.g.
Since then, more than 110 million infections and 2.5 million deaths have been reported.
Because the situation remains dynamic, please replace with: by (date), more than 110 m infections --- etc.
women are prudent to be speculated as a potential susceptible group
Not sure what this means. It seems that the reference is made to initial fears about the risk to pregnant women.. perhaps because of previous experience with H1N1. Please revise the sentence
Results: please insert correct reference.
Results: please note that the issue of vaccination remains controversial because of the lack of scientific studies in pregnant women. The views expressed by reference 14 are those of the authors and are not shared widely by regulatory bodies. You may wish to consider that there is a Place for vaccination in high risk individuals.
Initial assessment: “shielding” was advocated at the start of the pandemic for the medically high risk obstetric patients e.g. those with solid organ transplant, congential cardiac disease..etc. You may wish to consider adding a reference to these measures esp because vaccines are unlikely to be universally available for some time, and there is likely to be some resistance to vaccination in pregnancy for those considered to be of low risk of mortality/morbidity from covid (compared to H1N1 which had a far more serious effect in pregnancy).
3.1.1 infection screening:
This section would benefit from revision to take into consideration the following:
- Infection screening is not the same as preadmission testing.
- Testing can be done for asymptomatic or symptomatic women.
- The type of test (rapid e.g. lateral flow or lab based testing) each has its drawbacks.
- Testing may be done for those who require admission e.g. labouring women, or for those who attend for antenatal care episodes.
It would be helpful to address these items separately.
Please also note that health care structures vary widely: not all test positive women are admitted to hospital. In addition, testing generates three outcomes: positive (with a possibility of false positive), negative (with a possibility of false negative) and those whose status is unknow either because they decline testing or if the test is awaited.
- Please also comment on the place of partners>
- Please also note that the availability of contact tracing varies widely.
3.1.2 risk factors also include ethnic minority and SE status. Please note that the majority of positive women do not require hospital admission (at least in the UK) – they are managed at home- a recent introduction is home pulse oximetry for home use.
3.1.3.1. the recommendation that elective ultrasound should not be performed is probably at variance with most guidance. As ultrasound is important in pregnancy – some scans may be reduced if judged to e redundant, but the rest are usually protected through the use of PPE, ventilation and social distancing. You may however be referring to ‘unnecessary’ rather than ‘routine”.
3.2.1. Instructing women to stay at home for two weeks – does not provide sufficient guidance to their interaction with family members. There is a description of preadmission or upon admission testing, and about the use of ventilation in rooms. But, there is insufficient analysis of the pros and cons, and insufficient consideration that the facilities described are unlikely to be available in the majority of settings.
The reference to CDC and WHO recommendations also needs to take into consideration that these have changed overtime and that availability is a major constraint.
The authors need to explore whether they regard the distinction between aerosolization and non-aerosol generating procedures as being still relevant in light of more informed knowledge about viral transmission.
Intrapartum management: the authors need to clarify whether this applies to all pregnant women or only those who are covid positive / symptomatic. Also, they need to update the section on vertical transmission as the incidence and affection in the fetus is very small.
Round 2
Reviewer 3 Report
Article addressed main issues raised in the review.